# Discussion on the Discreteness of the Attenuation Parameters of the Peak Particle Velocity Induced by Blasting

**DOI:** 10.3390/s24051355

**Published:** 2024-02-20

**Authors:** Zhaowei Yang, Yingguo Hu, Meishan Liu, Peng Li, Erlei Yao, Chenyang Ma

**Affiliations:** Key Laboratory of Geotechnical Mechanics and Engineering of Ministry of Water Resources, Changjiang River Scientific Research Institute, Wuhan 430010, China; yangzw@whu.edu.cn (Z.Y.);

**Keywords:** blast seismic wave, peak value of vibration, propagation attenuation, attenuation coefficient, discreteness

## Abstract

The research on the attenuation law of blasting vibration has become the foundation and precondition of the effective control of blasting vibration damage. Aiming at the characteristics of low frequency, low velocity, and strong amplitude of the R wave, an improved wave component separation method based on R wave suppression is proposed. Combined with the measured vibration signals of a field test, the attenuation parameters of different types of waves in the propagation process of blasting seismic waves are studied. The analysis results show that, in the process of blasting seismic wave propagation, the attenuation parameters of different types of waves are significantly different. With an increase in propagation distance, the proportion of the different types of waves will also change. The study of attenuation law with only coupled particle peak vibration velocity often showed high discreteness. The fitting correlation coefficient and prediction accuracy of peak vibration velocity without distinguishing wave modes are lower than those induced by the P wave or R wave alone, which should be attributed to the conversion of dominant wave modes in blasting vibration at different distances.

## 1. Introduction

Because of its high efficiency and low cost, blasting excavation has become an essential engineering excavation method. It is more frequently being used in national economic construction, such as water conservancy, road construction, mining and national defense. However, due to the variability of blasting operation environments and the incompleteness of blasting theory, blasting operations will inevitably have a certain negative impact. For example, the propagation of the blasting seismic wave will lead to the vibration of adjacent slopes and buildings (structures) and affect their stability. When the induced vibration exceeds a certain threshold, it may lead to engineering safety problems, such as a slope landslide, an engineering collapse and even casualties [1,2]. Therefore, the research on the propagation and attenuation law of the blasting seismic wave is called one of the hot topics in the field of blasting engineering.

For a long time, the research on the attenuation law of blasting particle peak vibration velocity has been widely concerned by scholars at home and abroad. Existing studies show that the single shot charge, propagation path, initiation mode and other factors have a great impact on vibration attenuation [3,4]. Wang Yujie et al. [5] calculated the seismic wave propagation attenuation coefficient of the rock mass of the underground powerhouse of Zhouning Hydropower Station by using the regression analysis method through the analysis of the measured vibration data of the presplitting blasting excavation of the protective layer and gave the propagation attenuation law of the blasting seismic wave in the intact granite body on this basis. When using the Sadovsky formula to study the attenuation law of rock blasting vibration, Hu Jianhua et al. [6] comprehensively considered the influence of the elevation difference between the particle and the blasting source and proposed an improved multivariate linear calculation model of the attenuation law of blasting vibration, which greatly improved the reliability of the attenuation calculation formula. Havenith et al. [7] showed that there is an obvious nonlinear relationship between the elevation amplification effect of vibration and the local variation in medium strain. Based on the elastic wave theory, Lu Wenbo et al. [8] proposed an improved attenuation formula of particle peak vibration velocity and demonstrated the effectiveness of the formula with field measured data and numerical simulation. Gao, Li and others [9,10,11,12] analyzed the propagation and attenuation law of blasting vibration through theoretical derivation or a field test.

The existing research shows that most of the research on the attenuation law of blasting vibration at this stage tends to take the peak vibration degree of coupled mass points as the main research object, but in the process of seismic wave propagation in rock mass medium with an increase in propagation distance, the proportion of the different types of waves in the blasting vibration will also change, and the dominant wave may change to another type. So, the attenuation law research with only the coupled particle peak vibration velocity will often have great discreteness [13,14]. Therefore, how to determine the attenuation law of the different types of waves is a key problem in the research of blasting vibration prediction.

Based on the research of the propagation mechanism of the blasting seismic wave combined with the field blasting test of the Fengning pumped storage power station, this paper discusses and studies the discreteness of the attenuation law of blasting particle peak vibration velocity so as to provide theoretical guidance for later engineering blasting design and the prediction of blasting induced vibration.

## 2. Composition and Evolution of the Blasting Seismic Wave

In the process of blasting construction, in addition to the energy released by explosive explosion for rock-mass crushing, part of the energy will spread around in the form of a seismic wave, resulting in the vibration of ground particles. Existing studies show that seismic waves can be divided into body waves and surface waves in the process of propagation, and body waves can be divided into P (longitudinal) and S (transverse) waves. According to the elastic wave theory [15], the propagation velocities of the P wave and S wave can be obtained with Equation (1):(1)$$\left.\begin{array}{c} V_{P}=\sqrt{\frac{\lambda +2G}{\rho }}\\ V_{S}=\sqrt{\frac{G}{\rho }} \end{array}\right\}$$
where λ, G are lame coefficients.

With an increase in propagation distance, the body wave quickly decreases to a lower level due to its rapid attenuation. At this time, the surface wave plays a dominant role in blasting vibration. Therefore, only the propagation characteristics of the surface wave in the medium can be used in a large range of the explosion center distance. As we all know, the surface wave (R wave) is a wave formed by the interference of the non-uniform P wave and non-uniform S wave under certain propagation conditions. Taking the wave propagation in a uniform elastic medium as an example, the Rayleigh surface wave equation can be expressed as follows:(2)$${\left(2-\frac{V_{R}^{2}}{V_{S}^{2}}\right)}^{2}-4\sqrt{1-\frac{V_{S}^{2}}{V_{P}^{2}}}\sqrt{1-\frac{V_{R}^{2}}{V_{S}^{2}}}=0$$
where $V_{R}$ is the velocity of the R wave; and E, μ and ρ are the dynamic elastic modulus, dynamic Poisson’s ratio and rock mass density of the rock mass transmission medium, respectively.

By solving Equations (1) and (2), the relationship between the longitudinal wave (P wave), transverse wave (S wave), Rayleigh surface wave (R wave) and Poisson’s ratio can be obtained. The relative relationship between the three is shown in Figure 1.
(3)$$\left\{\begin{array}{c} \frac{V_{P}}{V_{S}}=\sqrt{\frac{2(1-\mu )}{1-2\mu }}\\ \frac{V_{R}}{V_{S}}=\frac{0.87+1.12\mu }{1+\mu } \end{array}\right.$$

According to the existing research, when the Rayleigh surface wave propagates in the medium, the horizontal displacement and vertical displacement equations of the medium pointing are as follows:(4)$$\left.\begin{array}{r} u_{x}=A(\frac{\omega }{V_{R}}e^{-\frac{2\pi }{\lambda_{R}}rz}-2cse^{-\frac{2\pi }{\lambda_{R}}sz})\sin(\omega (t-\frac{x}{V_{R}}))\\ u_{z}=A(\frac{2c\omega }{V_{R}}e^{-\frac{2\pi }{\lambda_{R}}sz}-re^{-\frac{2\pi }{\lambda_{R}}rz})\cos(\omega (t-\frac{x}{V_{R}})) \end{array}\right\}$$
where, $c=\frac{\sqrt{1-pn}}{2-p}$, $\lambda_{R}=\frac{2\pi V_{R}}{\omega }$, $p={\left(\frac{V_{R}}{V_{S}}\right)}^{2}$, $n={\left(\frac{V_{S}}{V_{p}}\right)}^{2}$, $r=\frac{\omega }{V_{R}}\sqrt{1-{\left(\frac{V_{R}}{V_{p}}\right)}^{2}}$, $s=\frac{\omega }{V_{R}}\sqrt{1-{\left(\frac{V_{R}}{V_{S}}\right)}^{2}}$.

When the rock mass medium is a Poisson body (μ=0.25), it can be calculated from Equations (4) and (5):(5)$$\left.\begin{array}{c} u_{x\left|z=0\right.}=0.4246Ak_{R}\sin(\omega t)\\ u_{z\left|z=0\right.}=0.6213Ak_{R}\cos(\omega t) \end{array}\right\}$$
where *A* is the peak ground displacement; and $k_{R}$ is the wave number of the R wave.

According to Formula (5), near the free surface of the rock mass, the particle motion trajectory of the R wave is elliptical, the ratio of the horizontal axis to the vertical axis of particle motion is approximately 2:3, and the vertical displacement phase is ahead of the horizontal displacement phase $\frac{\pi }{2}$.

When the rock mass medium is a Poisson body, the variation in horizontal displacement and vertical displacement with amplitude depth can be calculated through Equation (4), as shown in Figure 2.

From the above analysis, we can obtain several characteristics of the Rayleigh surface wave as the theoretical basis for surface wave recognition and suppression:(1)In the same rock mass medium propagation process, the propagation speed of the longitudinal wave (P wave) is the fastest, followed by the shear wave (S wave), and the surface wave (R wave) is the slowest.(2)The surface wave propagation velocity and shear wave velocity are approximately linear in numerical value, and with an increase in the Poisson’s ratio, they are approximately close in numerical value. Therefore, the shear wave velocity can be approximately determined in the later R wave region.(3)It can be observed from Equations (4) and (5) that the particle trajectory is elliptical due to the propagation of the surface wave, which can be used as a further discrimination after the final identification of the surface wave.

## 3. Attenuation Calculation of Peak Value of Blasting Vibration Velocity

In the process of rock blasting excavation, in addition to the energy released by explosive explosion for rock mass crushing, some of it will spread outward in the form of a blasting seismic wave, resulting in vibration of the rock mass medium. For the problem of seismic wave propagation attenuation, researchers at home and abroad have discussed it in detail and put forward different vibration attenuation formulas. In China, the Sadovsky formula of the former Soviet Union is usually used to describe the attenuation relationship between vibration peak and explosion center distance [16,17], and the expression is as follows:(6)v=k(Q1/3R)α
where v is the particle vibration velocity; k is the site coefficient, and α is the attenuation index, both of which are related to the blasting scheme and geological conditions; *Q* is the maximum charge weight per delay interval; and *R* is the distance between the explosion source and measuring point.

It can be observed from Equation (6) that, under the premise of a known blasting design, the attenuation law of blasting vibration propagation in a rock mass medium can be determined with the measured vibration data.

The logarithmic treatment of both sides of Equation (7) can obtain the following:(7)$$\mathrm{lg}v=\mathrm{lg}k+\alpha \mathrm{lg}(\frac{Q^{1/3}}{R})$$

Set y=lgv, $x=\mathrm{lg}(\frac{Q^{1/3}}{R})$, b=lgk, Equation (7) can be transformed into a linear equation as follows:(8)y=αx+b

Therefore, the sum of the squares of errors between all linear regression values and actual monitoring values can be expressed as follows:(9)$$f(\alpha ,b)={\sum \left(y_{0}-y_{i}\right)}^{2}={\sum \left(\alpha x_{i}+b-y_{i}\right)}^{2}$$

According to the principle of the least square method, the attenuation index α, b can be calculated as follows:(10)$$\left.\begin{array}{c} \alpha =\frac{\sum \left(x_{i}-\hat{x})(y_{i}-\hat{y}\right)}{\sum {\left(x_{i}-\hat{x}\right)}^{2}}\\ b=\hat{y}-\alpha \hat{x} \end{array}\right\}$$

In the process of linear analysis, the correlation coefficient COR is often used to evaluate the fitting degree of the linear regression line. In this paper, the correlation coefficient COR can be used to describe the correlation between the independent variable x and dependent variable y in linear Equation (8), that is, the effectiveness of blasting vibration peak prediction under different blasting center distance r, which can be expressed as follows:(11)$$COR^{2}=\frac{\sum \left(x_{i}-\hat{x})(y_{i}-\hat{y}\right)}{\sum {\left(x_{i}-x\right)}^{2}\sum {\left(y_{i}-y\right)}^{2}}$$

## 4. Attenuation Law of Particle Peak Vibration Velocity of Blasting Coupling Vibration

In order to have a more direct and in-depth understanding of the attenuation law of particle peak vibration velocity induced by blasting seismic wave propagation, the attenuation characteristics of the seismic wave are analyzed in detail by selecting the measured vibration data of a vertical hole blasting test of the Fengning pumped storage power station.

The Fengning pumped storage power station is located in Fengning Manchu Autonomous County, Hebei Province, with an installed capacity of 3.6 million KW. The power station is excavated and constructed in two phases. It is the largest pumped storage power station in China. In order to explore the propagation and attenuation law of the blasting seismic wave, a field blasting test was carried out in the second-phase engineering geological exploration tunnel. Based on the topographic maps, the experimental sites could be considered flat. The rocks in the test area are mainly intact granite with high uniaxial compression strength and mechanical intensity. In this blasting test, six vertical blast holes are arranged, all of which are 76 mm in diameter. The holes are connected by half-second detonators and detonated one by one. The detailed hole network parameters are shown in Table 1. During the test, the blasting vibration intelligent monitor is used to monitor the surface vibration induced by the blasting. A total of six vibration monitoring points are arranged along the measuring line at a distance of 20–150 m from the blasting center, as shown in Figure 3.

The vibration signals induced by blasting are measured with a blasting vibration intelligent monitor called TC-4850 (Figure 4) during the in situ blast experiment. IBM portable computers with BM View analysis software (Blasting Vibration Analysis) were used to withdraw field test data. The measurement amplitude range and frequency range of TC-4850 are 0.001–35.4 cm/s and 1–500 Hz, respectively, which can fully cover all ranges required for blasting vibration without the need for additional ranges.

Taking the 10# measuring point as an example, Figure 5 shows the measured vertical time history curve of typical blasting vibration. From the figure, it can be observed that the vibration waveform is obviously divided into six sections, which are produced by the blasting of six blast holes, respectively. The peak vibration velocity of each measuring point according to the measured vibration waveform is read. Since the rock mass at the 1# measuring point nearest to the explosion source is relatively broken, the data of the measuring point are excluded from the analysis.

Figure 6 is a vibration attenuation formula linearly fitted according to Formula (9) and its fitted correlation coefficient.

## 5. Attenuation Law of Particle Peak Vibration Velocity Based on Separating P Wave and R Wave

### 5.1. R Wave Separation of Measured Vibration Signals

Because of its low frequency, long duration and large carrying energy, the surface wave is the most destructive waveform in blasting vibration data. At the same time, in the actual vibration monitoring data, because the surface wave is interleaved in the reflected wave area, the signal-to-noise ratio of the vibration signal is low, which may further destroy the basic characteristics of the P- and S-wave signals in the vibration signal. Therefore, when accurately analyzing the measured vibration signal, first extract and suppress the surface wave signal, which is conducive to improve the signal-to-noise ratio of the measured vibration and provides effective help for the further analysis of the propagation and attenuation of subsequent blasting seismic wave in the rock mass.

Surface wave removal from seismic signals is no longer a new problem in geophysical research. At present, there are some mature surface wave extraction and suppression methods, such as F-x filtering, F-k filtering, wavelet transform and so on [18,19,20]. However, the research ideas of these methods are basically the same: by transforming the seismic signal from the space–time domain to other processing domains, such as the frequency wavenumber domain and frequency space domain, the difference between the surface wave and body wave (P wave and S wave) is analyzed, the surface wave is extracted and suppressed.

There is an obvious disadvantage in the existing methods, that is, the protection of the body wave signal is not considered in the process of surface wave suppression; therefore, it may cause further damage to the body wave signal, which does not meet the requirements of “high precision and high fidelity” in seismic signal processing.

#### 5.1.1. Surface Wave Recognition Method Based on S-Transform

In the process of blasting seismic wave attenuation law and vibration prediction, because the energy carried by the surface wave accounts for the main part, suppressing and identifying the surface wave are very important tasks in blasting vibration signal processing, which play a very important role in improving the accuracy of blasting impact evaluation. According to the characteristics of the surface wave, such as low frequency, large amplitude, low propagation speed and long vibration duration, scholars at home and abroad have proposed a variety of methods to identify and suppress the surface wave, such as low-frequency filtering and high-frequency filtering, frequency wavenumber domain (f-k domain) filtering, wavelet transform and so on. Each method has a certain effect on the recognition and suppression of the surface wave, but there are some limitations. For example, when suppressing the surface wave in the whole-time domain, the low-frequency part of the body wave will also be removed. In this section, in order to effectively suppress the wave interference on the measured blasting vibration signal without disturbing the body wave information, a method of suppressing the surface wave information with S-transform is proposed. This method inherits and develops the advantages of short-time Fourier transform and wavelet transform. Its window width decreases with an increase in signal frequency, it has high resolution for high-frequency signals, and it can flexibly use different filters according to the characteristic distribution of the frequency of the wave above the vibration signal at different times, so as to effectively protect the frequency component of the body wave signal, which has high practical significance.

#### 5.1.2. S-Transform

S-transform is a combination of STFT and CWT and is an effective time–frequency analysis method for non-stationary signals. To better control the resolution of the S-transform, the S-transform that is fine-tuned with parameter p is given as follows:(12)$$S(\tau ,f)=\frac{\left|f\right|}{p\sqrt{2\pi }}\int x(t)e^{-\frac{{\left(t-\tau \right)}^{2}f^{2}}{2p^{2}}}e^{-2i\pi ft}dt$$

As S-transform is directly related to the Fourier transform as STFT, it can be calculated with fast Fourier transform (FFT) and inverse fast Fourier transform (IFFT), which can provide a low computation complexity of S-transform. On the other hand, in the view of CWT, the S-transform can be rewritten as follows:(13)$$S(\tau ,f)=e^{-i2\pi ft}\sqrt{\left|f\right|}W(\tau ,a)$$
where a represents the scale inversely proportional to frequency, and *W*(*τ*, *a*) is the CWT of the signal *x*(*t*) with a special complex Morlet wavelet satisfying the following equation:(14)$$\Phi (t)=\frac{1}{p\sqrt{2\pi }}e^{-\frac{t^{2}}{2p^{2}}}e^{-i2\pi t}$$

To obtain the energy distribution of the signal in the time–frequency domain, the squared modulus of the S-transform is taken into account. The S-transform spectrogram is given as follows:(15)$${\left|S(\tau ,f)\right|}^{2}=S(\tau ,f)S^{*}(\tau ,f)$$

The gaussian window of S-transform could provide a higher time resolution for high frequencies and a higher frequency resolution for low frequencies. The seizure EEG signals contain high-frequency components, such as spike waves and sharp waves, while the frequency of normal EEG signals is relatively low; therefore, the S-transform spectrogram is an effective method to characterize seizure EEG signals. The parameter p in the Gaussian window allows for the adjustment of the energy concentration. After sufficient experiments, the value of p is set to 0.5 to obtain better EEG features.

#### 5.1.3. Filter Design

The time–frequency domain filtering process of blasting vibration can be expressed with the following formula:(16)$$R(t)=ST^{-1}[ST[x(t)]\cdot N(t,f)]$$

According to the surface wave characteristics, if the distribution area of the surface wave signal on the time–frequency domain diagram is $D_{R}$, the frequency filter function can be expressed as follows:(17)$$N(t,f)=\left\{\begin{array}{c} 1\begin{array}{cc} & (t,f)\in D_{R} \end{array}\\ 0\begin{array}{cc} & (t,f)\notin D_{R} \end{array} \end{array}\right.$$
where f is the frequency of the R wave.

### 5.2. Study on Attenuation Law of P Wave and R Wave

#### 5.2.1. Study on Attenuation Law of P Wave

As we all know, compared with the S wave and R wave, the P wave propagates faster. Therefore, generally, in the vibration waveform monitored with the sensor, the P wave always arrives first, followed by the S wave, R wave and other continuous waveforms. As for the first break of the S wave, the author introduced it in detail in reference [21]. In order to avoid repetition, the text directly quoted its recognition method on which the attenuation of the P wave was studied. Therefore, it can be observed that, if these successive waves do not arrive before the first to maximum peak of the vibration waveform, the corresponding vibration peak is caused by the P wave, as shown in Figure 7. The attenuation law of the vibration peak induced by the P wave at each monitoring point is fitted and analyzed, and the fitting results are shown in Figure 8.

#### 5.2.2. Study on Attenuation Law of R Wave

Using the R wave separation method described above to analyze and process the measured vibration signal of the Fengning field test combined with filter analysis, the vibration component caused only by the R wave can be further obtained. The separation results are shown in Figure 9.

The R wave attenuation law fitting analysis is carried out on the measured vibration data of each monitoring point, and the fitting results are shown in Figure 10.

## 6. Analysis and Discussion of Experimental Data

From Figure 8, Figure 9 and Figure 10, the attenuation formula of the particle peak value of blasting coupling vibration and the particle vibration peak value, respectively induced by the separated P wave and R wave, through Equation (9) are compared in Table 2.

It can be observed from Table 2 that there are great differences in the fitting attenuation parameters and correlation coefficients of the three types of vibration peaks. First, compared with the particle peak vibration attenuation fitting of coupled vibration, the correlation coefficient of the particle vibration attenuation fitting formula based on the separation of the P and R waves is significantly improved. At the same time, there are obvious differences in the attenuation coefficients of the three types of peak vibration velocity. The attenuation coefficient K that is fitted by P-wave induced vibration is higher than the attenuation parameters of the coupled vibration peak, while the attenuation coefficient that is fitted by R wave induced vibration is significantly lower. The value of K is only 20–50% of the coupled vibration and 10–30% of the P-wave induced vibration; Compared with the coupled vibration, the reduction rate of a value is 5–20% (average reduction rate is 15.2%), which is 12–30% (average reduction rate is 22.5%) lower than that of P-wave induced vibration. Therefore, it can be concluded that the attenuation laws fitted by the different types of particle vibration peaks are quite different in the propagation process of the blasting seismic wave in a rock mass medium, and the vibration attenuation induced by the R wave is the slowest. Therefore, it can be roughly summarized as follows: with an increase in propagation distance, the vibration prediction using the attenuation formula obtained by coupling the measured vibration signal with the vibration peak fitting will inevitably produce large errors, and the type of wave dominating the peak vibration velocity will also change with an increase in propagation distance.

In order to quantitatively analyze the relationship between propagation distance and dominant type waves, the following prediction formula is established:(18)kP(Q1/3R)αP=kR(Q1/3R)αR

By substituting the attenuation parameters fitted in the table into Equation (19), it can be obtained that the main influence area of the R wave is $\frac{R}{Q^{1/3}}\geq (50\sim 60)$. Therefore, the attenuation formula of peak vibration velocity in the process of seismic wave propagation can be obtained as follows:(19)v=kP(Q1/3R)αPR<(50~60)Q1/3kR(Q1/3R)αRR≥(50~60)Q1/3

At the same time, the average error value a (the average error is the ratio of the sum of relative errors to the number of samples) generated by predicting the blasting vibration peak at each measured position based on the attenuation formulas of different pairs of vibration peak values is also given in the table. Comparing the data in the table, it can be observed that, compared with the measured values, the average error of using the attenuation formula of coupling peak vibration velocity fitting to predict the particle vibration peak is 29–40%, while the prediction error of using the attenuation formula based on P-wave and R wave fitting is 18–30% and 15–22%, respectively. It shows that using the attenuation formula based on P-wave and R wave induced particle vibration peak fitting proposed in this paper to predict the particle vibration velocity on the blasting surface can more accurately reflect the basic law of seismic wave attenuation with propagation distance, and the prediction is more accurate.

As was stated above, the characteristics of the blast source and the different attenuation along the propagation path jointly determine the composition and evolution of the wave. In the process of prediction and control of blasting vibration, it is necessary to analyze the composition of blasting seismic waves and evaluate the influence of different waves by combining the characteristics of blasting source, relative position of the measuring point and attenuation characteristics.

We merely analyzed the vertical velocity peak value’s attenuation parameter depending on the wave type. The influence of the source type and location in depth and the frequency dependency of attenuation parameters would be taken into consideration in a further study.

## 7. Conclusions and Suggestions

In view of the fact that the research on the attenuation law of particle peak vibration velocity in the engineering blasting field often only adopts the coupled vibration peak, but the vibration caused by various waves has different attenuation parameters, and the attenuation analysis using coupled vibration often leads to large errors, this paper proposes a method for attenuation research based on the particle peak vibration velocity induced by separated P and R waves combined with the fitting analysis of the measured data of a blasting field test. The following conclusions are drawn:(1)A calculation formula considering the peak attenuation of particle vibration induced by the P wave and R wave is proposed. Compared with the coupled peak attenuation formula, the fitting correlation coefficient of the formula is higher, and the particle vibration of vibration blasting is more accurate.(2)According to the fitting results of the measured vibration data, it can be concluded that the peak attenuation coefficient a of vibration induced by the R wave is 12–30% lower than that of the P wave. Therefore, with an increase in propagation distance, the vibration components induced by the different types of waves in the measured blasting vibration will also change.(3)An R wave suppression extraction method based on the measured blasting vibration signal is proposed. This method mainly realizes the extraction of the R wave according to the motion trajectory and time–frequency characteristics of R wave particles.

There are many factors affecting the attenuation of blasting vibration. This paper only preliminarily studies the influence of different kinds of waves on the attenuation of particle vibration peak and the discreteness of fitting. Of course, the existence of structural plane and crack will also affect the discreteness of the attenuation law. In the follow-up research, more engineering examples need to be combined to establish a model that comprehensively considers the influence of the different types of waves and structural planes on the vibration attenuation law.

## Figures and Tables

**Figure 1 sensors-24-01355-f001:**
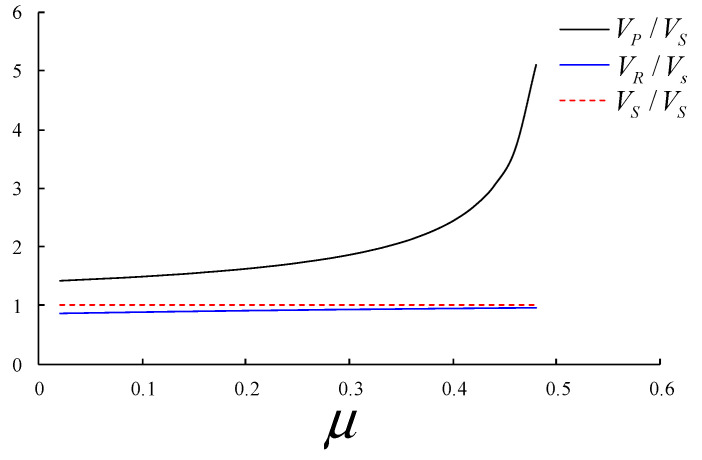
Sketch of the relationship between the wave velocities and μ.

**Figure 2 sensors-24-01355-f002:**
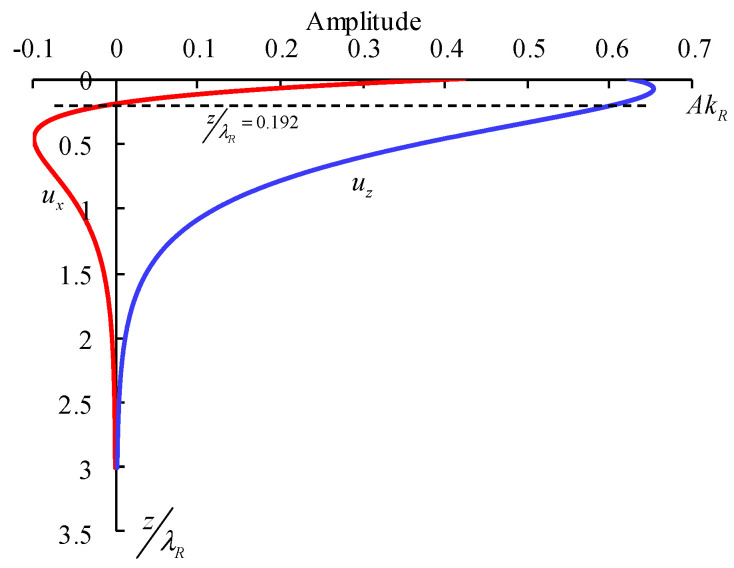
The variation curve of $u_{x}$, $u_{z}$ with depth.

**Figure 3 sensors-24-01355-f003:**
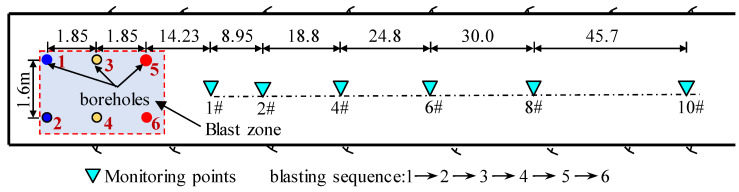
Arrangement of testing points and blasting holes.

**Figure 4 sensors-24-01355-f004:**
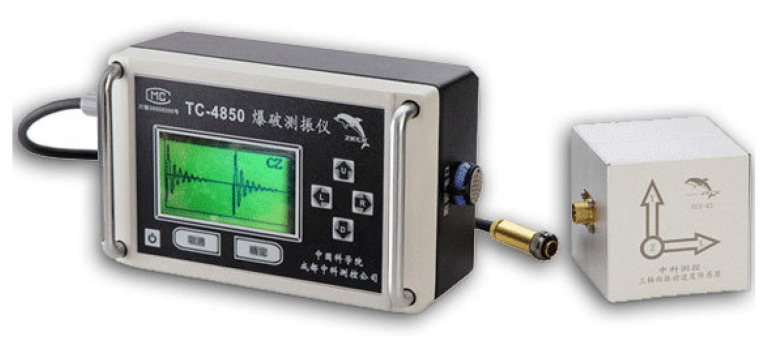
Blasting vibration intelligent monitor (TC-4850).

**Figure 5 sensors-24-01355-f005:**
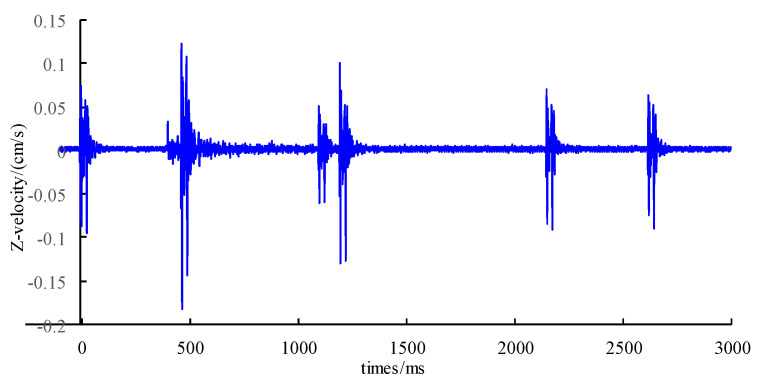
Measured blasting vibration velocity curves.

**Figure 6 sensors-24-01355-f006:**
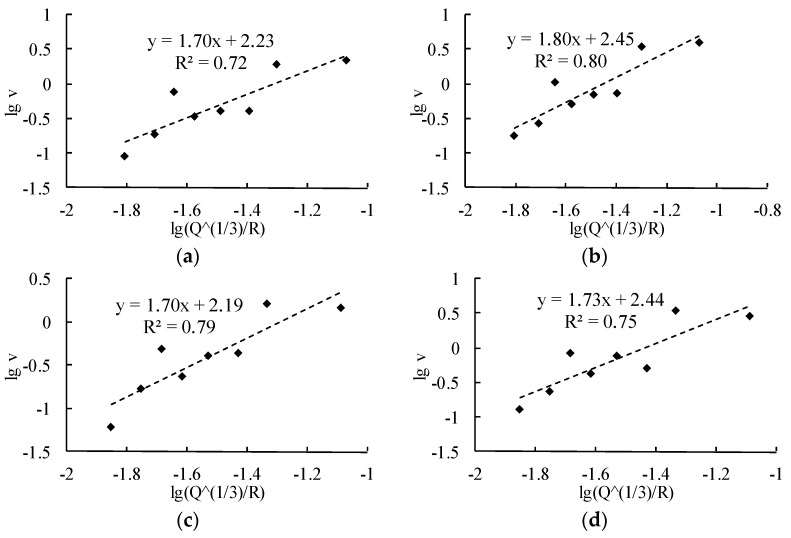
Fitting analysis of the coupled particle peak vibration velocity. (**a**) PPV induced by borehole 1. (**b**) PPV induced by borehole 2. (**c**) PPV induced by borehole 3. (**d**) PPV induced by borehole 4. (**e**) PPV induced by borehole 5. (**f**) PPV induced by borehole 6.

**Figure 7 sensors-24-01355-f007:**
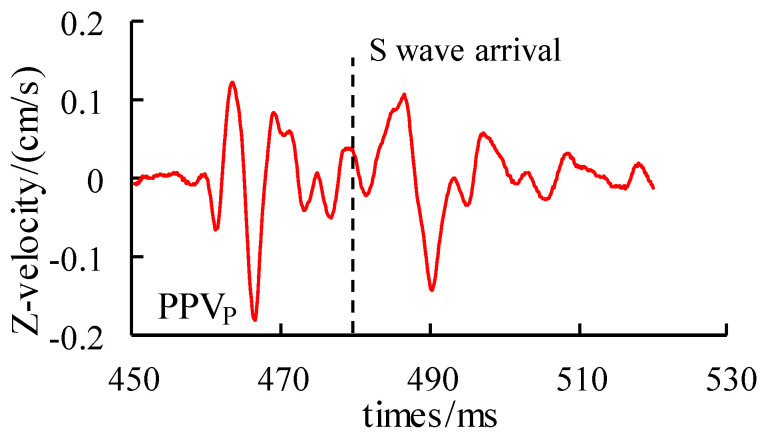
Typical waveforms of blasting vibration.

**Figure 8 sensors-24-01355-f008:**
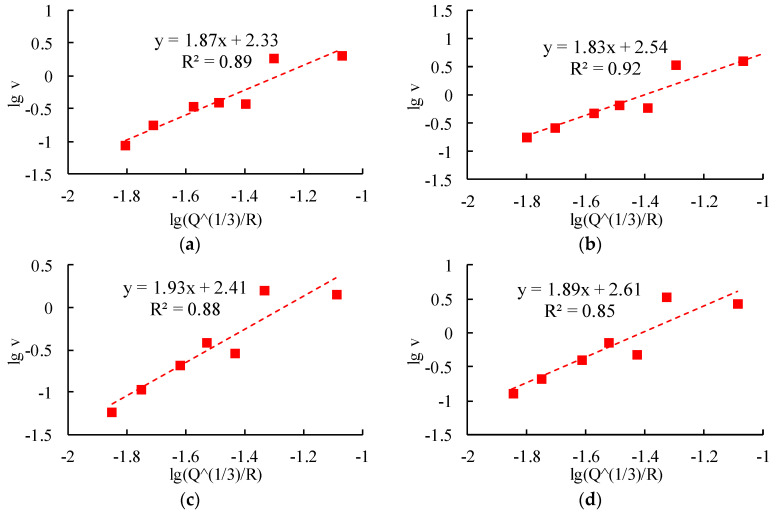
Fitting analysis of the vibration velocity of the P wave. (**a**) PPV of the P wave induced by borehole 1. (**b**) PPV of the P wave induced by borehole 2. (**c**) PPV of the P wave induced by borehole 3. (**d**) PPV of the P wave induced by borehole 4. (**e**) PPV of the P wave induced by borehole 5. (**f**) PPV of the P wave induced by borehole 6.

**Figure 9 sensors-24-01355-f009:**
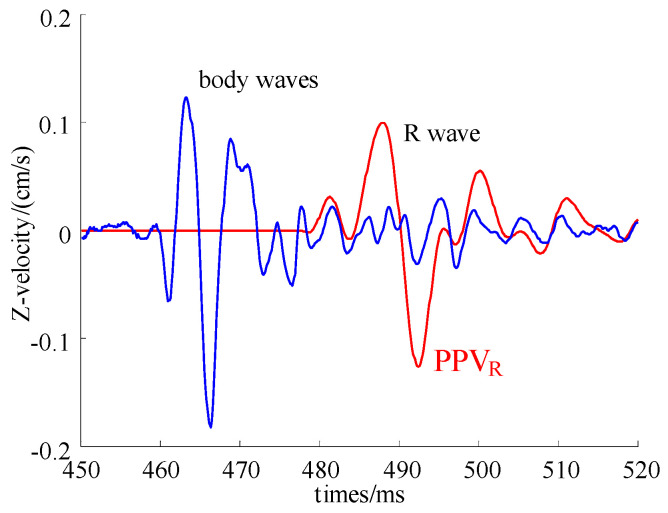
Typical waveforms of blasting vibration induced by the P and R waves.

**Figure 10 sensors-24-01355-f010:**
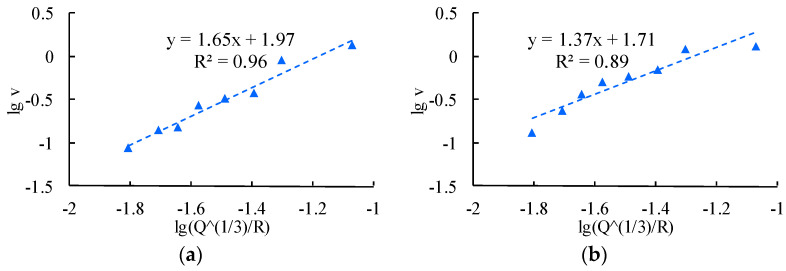
Fitting analysis of the vibration velocity of the R wave. (**a**) PPV of the R wave induced by borehole 1. (**b**) PPV of the R wave induced by borehole 2. (**c**) PPV of the R wave induced by borehole 3. (**d**) PPV of the R wave induced by borehole 4. (**e**) PPV of the R wave induced by borehole 5. (**f**) PPV of the R wave induced by borehole 6.

**Table 1 sensors-24-01355-t001:** Parameters of the blasting design of the field experiment.

	Blast Hole Diameter/mm	Hole Depth/cm	Diameter of Emulsion Explosives/mm	Charge Length/cm	Stemming Length/cm	Charge Weight per Delay Interval/kg
1	76	800	50	600	200	12.0
2	76	800	50	600	200	12.0
3	76	600	50	420	180	8.4
4	76	600	50	420	180	8.4
5	76	450	50	270	180	5.4
6	76	450	50	270	180	5.4

**Table 2 sensors-24-01355-t002:** Attenuation laws of PPV induced by different types of waves.

		PPV				${PPV}_{P}$				${PPV}_{R}$			
		K	α	$R^{2}$	ς	K	α	$R^{2}$	ς	K	α	$R^{2}$	ς
FN	1	169.82	1.70	0.72	29.2%	213.79	1.87	0.89	24.4%	93.33	1.65	0.96	14.8%
	2	281.84	1.81	0.80	40.5%	346.74	1.83	0.92	19.8%	51.29	1.37	0.89	22.3%
	3	154.88	1.74	0.79	32.2%	257.04	1.93	0.88	28.2%	25.70	1.39	0.91	22.6%
	4	275.42	1.73	0.75	37.3%	407.38	1.89	0.85	29.7%	50.12	1.43	0.91	18.3%
	5	169.82	1.61	0.77	37.4%	223.87	1.76	0.86	29.4%	46.77	1.41	0.89	25.4%
	6	288.40	1.73	0.84	39.4%	407.38	1.98	0.91	24.5%	56.23	1.45	0.91	21.2%

## Data Availability

Data will be made available on request.

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
