# Peer review of "Discussion on the Discreteness of the Attenuation Parameters of the Peak Particle Velocity Induced by Blasting"

_sensors, 2024, doi:10.3390/s24051355_

Round 1

Reviewer 1 Report

Comments and Suggestions for Authors

Dear Authors,

This paper aims to examine the distance-dependent attenuation of blast-induced vibrations, utilizing the Sadovsky formula's parameters: K (site coefficient) and alpha (attenuation index). The paper presents a theoretical background, covered in a previous publication by the same authors (reference 21), and applies the theory to measurements from blasting tests on a Chinese power station. The attenuation is assessed for three different peak particle velocities (PPV), the global one, one specific to the P-wave and one specific for the Rayleigh surface wave. To do so the authors are using a R-wave separation method described in a previous paper. The authors conclude that separating the two types of waves leads to better results by increasing the linear regression coefficient. They find thus different attenuation parameters for the two types of waves.

While the paper is well written and easy to understand, several serious concerns need to be addressed for publication.

1- Target Audience and References:

The article appears tailored to the Chinese scientific community, lacking a broader global perspective. References predominantly stem from the Chinese scientific community, while the topic has been studied worldwide. Consider referencing studies such as Athanasopoulos et al. (2000) or Kim and Lee (2000), which present more general laws derived from wave propagation theory instead of solely the Sadovsky formula.

2- Lack of Reference:

Include a reference for the development of the Sadovsky formula. For instance, cite the paper by Gao et al. (2022) in applied science to support the formula's background and evolution.

3- Repetition of Previously Published Material:

The theory presented in this paper replicates the content of the authors' previous work, including identical text and figures. Avoid redundancy by summarizing the theory briefly, referring readers to the previous publication for comprehensive details.

4- Comparison with Previous Studies and Geological/Geotechnical Settings:

Include a comparison with previous studies on the same subject to establish the novelty of the research. Furthermore, provide a description of the geological and geotechnical settings of the study site, which is currently missing.

4- Lack of Sensor Description:

Provide essential details about the sensors used in the study, such as their sensitivity, response characteristics, and any signal processing or filtering applied. Include information about the frequency content of the recorded signals and the maximum frequency captured.

5- Neglect of Horizontal Components:

Address the impact of horizontal components of blast-induced vibrations. Discuss the potential variation in energy ratio between the vertical and horizontal components of the R-wave with distance and its implications for attenuation measurements.

6- Frequency Dependency of Attenuation Parameters:

Investigate and discuss whether the estimated attenuation parameters are frequency-dependent. Relying solely on peak particle velocities (PPV) cannot lead to conclusive findings in this regard.

7- Differentiation of Wave Types and Geometric Effects:

Differentiate the geometric and inelastic terms contributing to seismic wave attenuation for each wave type. Acknowledge that both terms vary depending on the wave type considered and the source type and location in depth. Consult Kim and Lee (2000) for further insights.

8- Linking Attenuation Parameters to Wave Propagation:

Explain the implications of the obtained attenuation parameters for understanding seismic wave propagation phenomena. The discussion should emphasize how the findings contribute to the broader understanding of this field.

9- Figure Quality and Information:

Improve the quality of Figures 1 and 2, considering that they have been published elsewhere. Figure 1 requires clarification, as it mixes velocity ratios with shear wave velocities. Provide a clear indication of units and address the discrepancy in the velocity Vs being close to 1 regardless of mu.

10- Figure References and Content:

Ensure consistency between the text and figures. For instance, clarify the reference to point #9 in Figure 3, which is mentioned in the text but not visible. In Figure 4, consider including Fourier spectra associated with the vertical component for comprehensive analysis.

11- Surface Wave to Body Wave Ratio and Distance Variation:

Discuss the ratio of surface waves to body waves, particularly considering that the separation method employed identifies a major portion of the S-wave as an R-wave. Elaborate on whether this ratio varies with distance from the source, prompting further analysis and comments.

To enhance the paper's suitability for publication, address the concerns raised, broaden the scope of references, provide necessary details regarding sensors and measurements, offer comprehensive discussions on various aspects, and ensure clarity and consistency in figures and their references.

Author Response

Dear editors,

Thanks very much for your letter concerning our manuscript entitled “Discussion on the discreteness of the attenuation parameters of the peak particle velocity induced by blasting”. We would like to express our sincere appreciation for the reviewers’ careful reading and helpful comments on our manuscript. We have studied their comments carefully and have made the correction which we hope will meet with their approval. The main corrections and detailed responses to the reviewers’ comments are described below. We have marked the correction portions in yellow bars in the revised manuscript.

Thanks again for your time and letter.

Yours sincerely,

Zhaowei Yang

E-mail: yangzw@whu.edu.cn

Responses to Reviewer 1:

Comment 1: Target Audience and References:

The article appears tailored to the Chinese scientific community, lacking a broader global perspective. References predominantly stem from the Chinese scientific community, while the topic has been studied worldwide. Consider referencing studies such as Athanasopoulos et al. (2000) or Kim and Lee (2000), which present more general laws derived from wave propagation theory instead of solely the Sadovsky formula.

 Answer 1:

Thanks for your suggestion. According to the reviewer’s comment, we have made modification on introduction and references of this paper.

Comment 2: Lack of Reference:

Include a reference for the development of the Sadovsky formula. For instance, cite the paper by Gao et al. (2022) in applied science to support the formula's background and evolution.

  Answer 2:

Thanks for your suggestion. We are very sorry for our negligence. The references have been added in the revised manuscript.

Comment 3: Repetition of Previously Published Material:

The theory presented in this paper replicates the content of the authors' previous work, including identical text and figures. Avoid redundancy by summarizing the theory briefly, referring readers to the previous publication for comprehensive details.

 Answer 3:

Thanks for your suggestion. The repeated contents of the manuscript have been deleted.

Comment 4: Comparison with Previous Studies and Geological/Geotechnical Settings:

Include a comparison with previous studies on the same subject to establish the novelty of the research. Furthermore, provide a description of the geological and geotechnical settings of the study site, which is currently missing.

 Answer 4:

Thanks for your suggestion. The details about the geological and geotechnical settings of the study area, such as property of rock mass, nature of joints and number of joints have been provided in the revised manuscript.

Comment 5: Lack of Sensor Description:

Provide essential details about the sensors used in the study, such as their sensitivity, response characteristics, and any signal processing or filtering applied. Include information about the frequency content of the recorded signals and the maximum frequency captured.

 Answer 5:

The details of sensor description have been suppled in the revised manuscript, including the range of measurement and frequency response.

Comment 6: Neglect of Horizontal Components:

Address the impact of horizontal components of blast-induced vibrations. Discuss the potential variation in energy ratio between the vertical and horizontal components of the R-wave with distance and its implications for attenuation measurements.

 Answer 6:

Thanks for your good question. During the blasting test, blast vibration of three directions are measured. Due to the large effects of wave components, this paper made the vertical vibration be the subject investigated, discussed the discreteness of the attenuation parameters of the peak particle velocity induced by blasting. But in the further study, the horizontal components of measured vibration would be taken into consideration.

Comment 7: Frequency Dependency of Attenuation Parameters:

Investigate and discuss whether the estimated attenuation parameters are frequency-dependent. Relying solely on peak particle velocities (PPV) cannot lead to conclusive findings in this regard.

 Answer 7:

During the research, the peak velocity attenuation rule of blasting is discussed based on Sodev's empirical formula. the frequency dependency of attenuation parameters would be taken into consideration in the further study.

Comment 8: Differentiation of Wave Types and Geometric Effects:

Differentiate the geometric and inelastic terms contributing to seismic wave attenuation for each wave type. Acknowledge that both terms vary depending on the wave type considered and the source type and location in depth. Consult Kim and Lee (2000) for further insights.

 Answer 8:

Thanks for your good suggestion. The main content of this manuscript is about vertical velocity peak value's attenuation parameter depending on wave type. Therefor it’s not considered the influence of the source type and location in depth and the shortcomings above need to be improved by later research.

Comment 9: Linking Attenuation Parameters to Wave Propagation:

Explain the implications of the obtained attenuation parameters for understanding seismic wave propagation phenomena. The discussion should emphasize how the findings contribute to the broader understanding of this field.

 Answer 9:

Thanks for your suggestion. This study can contribute to understand the attenuation mechanism of vibration induced by blast wave and provide the possibility of predicting and controlling blasting vibration from the perspective of wave pattern analysis.

Comment 10: Figure Quality and Information:

Improve the quality of Figures 1 and 2, considering that they have been published elsewhere. Figure 1 requires clarification, as it mixes velocity ratios with shear wave velocities. Provide a clear indication of units and address the discrepancy in the velocity Vs being close to 1 regardless of mu.

 Answer 10:

The Fig.1 and Fig.2 have been revised in the manuscript.

Comment 11: Figure References and Content:

Ensure consistency between the text and figures. For instance, clarify the reference to point #9 in Figure 3, which is mentioned in the text but not visible. In Figure 4, consider including Fourier spectra associated with the vertical component for comprehensive analysis.

 Answer 11:

The Fig.3 and Fig.4 have been revised in the manuscript.

Comment 12: Surface Wave to Body Wave Ratio and Distance Variation:

Discuss the ratio of surface waves to body waves, particularly considering that the separation method employed identifies a major portion of the S-wave as an R-wave. Elaborate on whether this ratio varies with distance from the source, prompting further analysis and comments.

 Answer 12:

Thanks for your suggestion. For the blasting seismic wave, the S wave is only the dominant wave in the near region and R wave gradually grows and develops with the propagation distance increased, dominates the vertical vibration in middle and far field.

Reviewer 2 Report

Comments and Suggestions for Authors

The manuscript has some interest to readers since it analyzes the different parameters involved in the peak velocity equations. A few issues should be addressed before the work is ready for publication:

- Please only cite authors last names.

- Please give preference to citation of classical texts when dealing with general topics, such as elastic wave theory, without having to resort to obscure texts in the authors native language, such as [15]. This is just an example of the various instances that happens throughout the text.

- Please edit equations, properly using the equation editor, in particular regarding parentheses.

- Figure 5: please inform in the captions what each graph represents. Please indicate the corresponding number of each measuring point in the graph.

- Figure 6: please discuss about parameters in light of the literature. Is there anything particular about them?

- Pg. 7: please change the statement “As we all know…” to a less colloquial one, since it is not suitable for a technical paper.

- Figure 7: please inform in the captions what each graph represents.

- At least a minimum amount of information should be provided of the type of rock that is object of the work, including a discussion of the pertinence of the values found compared to those in the literature for similar rocks.

- Last sentence before section 7 (Conclusions): What is the word “Quickly” doing here?

- I do not believe the part in the title that says “Discussion on the discreteness…”. Please consider alternative to “discussion”. Also “discreteness” seems to appear to contrast with a continuum approach, which is not the case.

Comments on the Quality of English Language

The manuscript will benefit from a read-through by an English-proficient individual. 

Author Response

Dear editors,

Thanks very much for your letter concerning our manuscript entitled “Discussion on the discreteness of the attenuation parameters of the peak particle velocity induced by blasting”. We would like to express our sincere appreciation for the reviewers’ careful reading and helpful comments on our manuscript. We have studied their comments carefully and have made the correction which we hope will meet with their approval. The main corrections and detailed responses to the reviewers’ comments are described below. We have marked the correction portions in yellow bars in the revised manuscript.

Thanks again for your time and letter.

Yours sincerely,

Zhaowei Yang

E-mail: yangzw@whu.edu.cn

Responses to Reviewer 2:

Comment 1: Please only cite authors last names.

Answer 1:

The format of the reference list has been revised in the manuscript.

Comment 2: Please give preference to citation of classical texts when dealing with general topics, such as elastic wave theory, without having to resort to obscure texts in the authors native language, such as [15]. This is just an example of the various instances that happens throughout the text.

Answer 2:

The classical references have been cited in the revise manuscript.

Comment 3: Please edit equations, properly using the equation editor, in particular regarding parentheses.

Answer 3:

After checking the full text, we have modified the formulas, formats in the revised manuscript.

Comment 4: Figure 5: please inform in the captions what each graph represents. Please indicate the corresponding number of each measuring point in the graph.

Answer 4:

Thanks for your good suggestion. The captions of each graph in Fig.5 have been added in the revised manuscript.

Comment 5: Figure 6: please discuss about parameters in light of the literature. Is there anything particular about them?

Answer 5:

Fig.6 in this manuscript illustrates that the P wave velocity is faster than that of S wave from the measured blasting wave signal.

Comment 6: Pg. 7: please change the statement “As we all know…” to a less colloquial one, since it is not suitable for a technical paper.

Answer 6:

The colloquial expressions in this manuscript have been revised.

Comment 7: Figure 7: please inform in the captions what each graph represents.

Answer 7:

Thanks for your good suggestion. The captions of each graph have been added in the revised manuscript.

Comment 8: At least a minimum amount of information should be provided of the type of rock that is object of the work, including a discussion of the pertinence of the values found compared to those in the literature for similar rocks.

Answer 8:

The information of rock mass at the blasting test area have been provided.

Comment 9: Last sentence before section 7 (Conclusions): What is the word “Quickly” doing here?

Answer 9:

This is a statement that are not enough rigorous and errors. And the sentence has been rewritten in the revised manuscript.

Comment 10: I do not believe the part in the title that says “Discussion on the discreteness…”. Please consider alternative to “discussion”. Also “discreteness” seems to appear to contrast with a continuum approach, which is not the case.

Answer 10:

Thank you for your good question. It has been modified in the revised manuscript. And the discreteness in this paper represents that the attenuation parameter estimated from measured vibration signals based on Sodev's empirical formula have a low correlation.

Reviewer 3 Report

Comments and Suggestions for Authors

Thus, one of the most difficult tasks for mining enterprises, namely forecasting the decrease in the strength of structural elements of protected buildings and structures during blasting, is solved in terms of the stress concentration coefficient, the time of exceeding the long-term tensile strength and the rate of crack growth. It is shown that the existence of stress concentrators in the form of natural inhomogeneities or defects in building materials of building elements exposed to seismic explosion and air shock waves leads to the growth of cracks.  It is necessary to approach the existing problems and already solved tasks in more detail. To determine the surface energy, the crack distribution in the samples of some materials and the tensile strength of these materials are determined. The crack size distribution is used to calculate the effective crack length. However, the weakening of various elastic waves requires a more detailed and individual approach and consideration of the specifics of the terrain. Creating a model in this case has a purely mathematical and theoretical meaning. The article should be devoted to a specific area of rocks or purely laboratory effects. The novelty of the scientific article should be improved taking into account the binding to specific breeds or localities.

Author Response

Dear editors,

Thanks very much for your letter concerning our manuscript entitled “Discussion on the discreteness of the attenuation parameters of the peak particle velocity induced by blasting”. We would like to express our sincere appreciation for the reviewers’ careful reading and helpful comments on our manuscript. We have studied their comments carefully and have made the correction which we hope will meet with their approval. The main corrections and detailed responses to the reviewers’ comments are described below. We have marked the correction portions in yellow bars in the revised manuscript.

Thanks again for your time and letter.

Yours sincerely,

Zhaowei Yang

E-mail: yangzw@whu.edu.cn

Responses to Reviewer 3:

Comment 1: Thus, one of the most difficult tasks for mining enterprises, namely forecasting the decrease in the strength of structural elements of protected buildings and structures during blasting, is solved in terms of the stress concentration coefficient, the time of exceeding the long-term tensile strength and the rate of crack growth. It is shown that the existence of stress concentrators in the form of natural inhomogeneities or defects in building materials of building elements exposed to seismic explosion and air shock waves leads to the growth of cracks.  It is necessary to approach the existing problems and already solved tasks in more detail. To determine the surface energy, the crack distribution in the samples of some materials and the tensile strength of these materials are determined. The crack size distribution is used to calculate the effective crack length. However, the weakening of various elastic waves requires a more detailed and individual approach and consideration of the specifics of the terrain. Creating a model in this case has a purely mathematical and theoretical meaning. The article should be devoted to a specific area of rocks or purely laboratory effects. The novelty of the scientific article should be improved taking into account the binding to specific breeds or localities.

Answer 1:

Thank you for your good question. This problem is nothing to do with the research contents of the article.

Round 2

Reviewer 1 Report

Comments and Suggestions for Authors

Dear Authors,

According to me, you only provided answers to some of my comments in my first review. Here are the points that should be further addressed :

Comment 1 : Nothing changed except the addition of two references (Gao, Li).

Comment 2 : Paragraph 3 remains the same, no reference was added.

Comment 3 : I don’t see any changes in the manuscript concerning the redundancy of Author’s previous work.

Comment 4 : No comparison with previous studied has been added.

Comment 5 : Done

Comment 6 : « But in the further study, the horizontal components of measured vibration would be taken into consideration. » : This deserves to be mentioned in the discussion.

Comment 7 : « the frequency dependency of attenuation parameters would be taken into consideration in the further study. »  : This deserves to be mentioned in the discussion.

Comment 8 : « The main content of this manuscript is about vertical velocity peak value's attenuation parameter depending on wave type. Therefor it’s not considered the influence of the source type and location in depth and the shortcomings above need to be improved by later research. »  : This deserves to be mentioned in the discussion.

Comment 9 : No proper answer was given to this comment.

Comment 10 : Done

Comment 11 : Done

Comment 12 : Done

Comments on the Quality of English Language

There are still some minor grammatical errors in the text. A thorough review should be done to correct this grammatical errors.

Author Response

Comment 1: Nothing changed except the addition of two references (Gao, Li).

Answer 1:

Thanks for your suggestion. We are very sorry for our negligence. The references have been added in the revised manuscript.

[13] Athanasopoulos G A, Pelekis P C, Anagnostopoulos G A. Effect of soil stiffness in the attenuation of Rayleigh-wave motions from field measurements[J]. Soil Dynamics and Earthquake Engineering, 2000, 19(4): 277-288.

[14] Kim D S, Lee J S. Propagation and attenuation characteristics of various ground vibrations[J]. Soil dynamics and Earthquake engineering, 2000, 19(2): 115-126.

[17] Gao Y, Fu H, Rong X, et al. Ground-borne vibration model in the near field of tunnel blasting[J]. Applied Sciences, 2022, 13(1): 87.

Comment 2: Paragraph 3 remains the same, no reference was added.

Answer 2:

Thanks for your suggestion. According to the reviewer’s comment, we have made modification on introduction and references of this paper.

Comment 3: I don’t see any changes in the manuscript concerning the redundancy of Author’s previous work.

Answer 3:

Thanks for your suggestion. The repeated contents (5.1 R-wave separation of measured vibration signals) of the manuscript have been deleted.

Comment 4: No comparison with previous studied has been added.

Answer 4:

Thanks for your suggestion. It has been added in the discussion and it has been outlined in red for emphasis.

Comment 5: Done

Answer 5:

Thank you for approving our job

Comment 6: « But in the further study, the horizontal components of measured vibration would be taken into consideration. » : This deserves to be mentioned in the discussion.

Answer 6:

Thanks for your suggestion. It has been added in the discussion and it has been outlined in red for emphasis.

Comment 7: « the frequency dependency of attenuation parameters would be taken into consideration in the further study. »  : This deserves to be mentioned in the discussion.

Answer 7:

Thanks for your suggestion. It has been added in the discussion and it has been outlined in red for emphasis.

Comment 8: « The main content of this manuscript is about vertical velocity peak value's attenuation parameter depending on wave type. Therefor it’s not considered the influence of the source type and location in depth and the shortcomings above need to be improved by later research. »  : This deserves to be mentioned in the discussion.

Answer 8:

Thanks for your suggestion. It has been added in the discussion and it has been outlined in red for emphasis.

Comment 9: No proper answer was given to this comment.

Answer 9:

Thanks for your suggestion. This study can contribute to understand the attenuation mechanism of vibration induced by blast wave and provide the possibility of predicting and controlling blasting vibration from the perspective of wave pattern analysis.

Comment 10: Done

Answer 10:

Thank you for approving our job

Comment 11: Done

Answer 11:

Thank you for approving our job

Comment 12 : Done

Answer 12:

Thank you for approving our job
